# Gentle Versus Strong Touch Classification: Preliminary Results, Challenges, and Potentials

**DOI:** 10.3390/s20113033

**Published:** 2020-05-27

**Authors:** Soheil Keshmiri, Masahiro Shiomi, Hidenobu Sumioka, Takashi Minato, Hiroshi Ishiguro

**Affiliations:** 1Advanced Telecommunications Research Institute International (ATR), Kyoto 619-0237, Japan; m-shiomi@atr.jp (M.S.); sumioka@atr.jp (H.S.); minato@atr.jp (T.M.); ishiguro@sys.es.osaka-u.ac.jp (H.I.); 2Graduate School of Engineering Science, Osaka University, Osaka 565-0871, Japan

**Keywords:** physical interaction, touch classification, human–agent physical interaction

## Abstract

Touch plays a crucial role in humans’ nonverbal social and affective communication. It then comes as no surprise to observe a considerable effort that has been placed on devising methodologies for automated touch classification. For instance, such an ability allows for the use of smart touch sensors in such real-life application domains as socially-assistive robots and embodied telecommunication. In fact, touch classification literature represents an undeniably progressive result. However, these results are limited in two important ways. First, they are mostly based on overall (i.e., average) accuracy of different classifiers. As a result, they fall short in providing an insight on performance of these approaches as per different types of touch. Second, they do not consider the same type of touch with different level of strength (e.g., gentle versus strong touch). This is certainly an important factor that deserves investigating since the intensity of a touch can utterly transform its meaning (e.g., from an affectionate gesture to a sign of punishment). The current study provides a preliminary investigation of these shortcomings by considering the accuracy of a number of classifiers for both, within- (i.e., same type of touch with differing strengths) and between-touch (i.e., different types of touch) classifications. Our results help verify the strength and shortcoming of different machine learning algorithms for touch classification. They also highlight some of the challenges whose solution concepts can pave the path for integration of touch sensors in such application domains as human–robot interaction (HRI).

## 1. Introduction

Touch is one of the most basic and yet highly effective means of nonverbal communication for transfer of affective feelings and the emotions [1]. Research provides ample evidence for its positive therapeutic impact on individuals’ physical and mental stress [2]. Humans appear to be equipped with an innate capacity for comprehending the meaning that are entailed in a touch [1]. However, providing other agencies such as robots or other embodied communication media with this ability is rather nontrivial [3,4,5]. It is apparent that such an ability finds application in a broad real-life domains from socially-assistive robots [6] and robot therapy [7,8] to embodied telecommunication [9].

In this respect, a considerable effort has been placed on devising methodologies for automated touch classification. Among these approaches, machine learning has played a pivotal role. Jung et al. [10] applied Bayesian classifier and support vector machine (SVM) on a touch dataset that comprised fourteen different types of touch (grab, hit, massage, pat, pinch, poke, press, rub, scratch, slap, stroke, squeeze, tap, and tickle). They reported 54.0% and 53.0% overall accuracies by these classifiers, respectively. Silvera-Tawil et al. [11] used LogitBoost algorithm to obtain an average classification accuracy of 71.0% on nine touch gestures (pat, push, scratch, slap, stroke, tap, pull, squeeze, and no-touch). These touch gestures were performed on an artificial skin which was worn on a mannequin’s arm. Naya et al. [12] combined linear discriminant analysis (LDA) with K-nearest-neighbor (KNN) to achieve an overall classification accuracy of 87.0% on five touch gestures (pat, scratch, slap, stroke, and tickle). They used a 44 × 44 sensor grid for collecting their touch data. Nakajima et al. [9] utilized SVM in a scenario in which human subjects performed seven touch gestures (grasp, hug, press, punch, rub, slap, and no-touch) on a balloon interface. Their classifier was able to achieve an overall accuracy of 75.0% between multiple subjects. van Wingerden et al. [13] applied a single-layer neural network with 46 nodes in its hidden layer on the same dataset that was originally introduced by Jung et al. [10]. They reported a 54.0% classification accuracy on this dataset. Murray-Smith et al. [14] achieved 75.0% accuracy for overall classification of four touch gestures (stroke, scratch, rub, and tap). Their accuracy was based on an artificial neural network that was trained on 26,880 examples. However, the authors did not provide information about the number of users/participants that were included in their evaluation. Robinson et al. [15] devised a model (called Tapback) that detected when the users tapped on the back of their smartphone (without having to separate their phone from their ear). In their setting that included 36 individuals, the users tapped on the back of their phone once (772 instances), twice (301 samples), or thrice (220 instances) for a total number of 1293 instances. The authors claimed that their system achieved 93.0% (single tap), 78.0% (tapping twice), and 56.0% (three consecutive taps) accuracies. However, the authors did not provide any details on training procedure/setting of their system. Braun et al. [16] used two microphones along with a support vector classifier for detecting four touch gestures on a surface (swipe, tap, knock, and stomp). They used 1247 samples from 13 participants and achieved overall accuracies between 91.0–99.0% and 97.0–100.0% using a single and double microphones, respectively. However, they utilized a large number of features (587 features) and their results appeared to be solely based on the training set. Harrison et al. [17] introduced a system called TapSense that allowed for detection of tip and tap with a pen, and nail and knuckle interactions with a finger. They used data from 18 participants that interacted with a multitouch Table (320 instances) and reported an overall accuracy of 88.30%. Finally, Alonso-Martin et al. [18] used contact microphones for classification of four touch gestures (stroke, tap, slap, and tickle) on a robot. Their study included 25 participants. The authors reported that the logistic model tree (LMT) achieved the best performance, with an F-score of 0.81. Although the authors did not explicitly report the accuracy of their model, their article included the LMT’s confusion matrix. The authors also provided a comprehensive review of different sensor technologies for touch interaction. Fleh et al. [19] present a comprehensive review on this topic.

Looking at the touch classification literature, one can undeniably appreciate substantial efforts with progressive results. At the same time, it also pinpoints two primary limitations in these studies. First, the reported results by these studies are mostly based on overall classification accuracies (but also see [18,20]). As a result, they do not provide an insight about the performance of their classifiers on each type of touch. For instance, by looking at these results, it is not possible to tell whether the reported overall accuracies are based on classifiers’ ability to predict all different types of touch or just a small subset of these types. The importance of such an ability becomes clearer once one considers the substantially reduced accuracy of such classifiers when their performance are examined as per different types of touch. For instance, Stiehl and Breazeal [7] used a three-hidden-layer neural network (100 nodes per layer) for classification of nine different types of touch (contact, pat, pet, poke, rub, scratch, slap, squeeze, and tickle) that were performed on a pet-robot’s arm. The authors noted that their model was not able to classify such touch gestures as slap due to its short duration. Furthermore, they reported that although their model generated a high true negative (e.g., rub and poke are not the same) for the remaining eight touch gestures (94.0–98.0%), it produced a highly varying true positive (i.e., predicting rub, pat, etc. with a high accuracy) results (23.0–84.0%).

Second, these results did not consider the same touch types that might only vary in their level of strength. For instance, they did not examine how well their classifiers could differentiate between a gentle and a stronger stroke. This is certainly a desirable ability since a change in intensity of a touch may utterly transform its meaning. For instance, whereas a gentle squeeze might signal an affectionate treat, a stronger one could be construed as a sign of punishment by its recipient.

The current study attempts to address these shortcomings by considering the accuracy of a number of classifiers for both, within- (i.e., same type of touch with differing strengths) and between-touch (i.e., different types of touch) classification. For this purpose, we wore a sensor-grid vest on a mannequin’s upper body and asked human subjects to perform different types of touch on it. We also asked these participants to perform these touch gestures in two different strengths: a gentle touch and a stronger touch.

The contributions of the present study are threefold. First, it introduces a low dimensional feature space whose computation relies on a very short time window of a touch trial. As a result, this feature space bypasses such issues as inability for classifying a touch gesture due to its short duration [7] or utilization of a large number of features [16]. Second, it provides a comparative analysis of six different classifiers on three types of touch, each with two different strengths: a gentle and a stronger touch (i.e., six different touch scenarios in total). This provides a preliminary results on the utility of these algorithms for classification of the touch gestures that differed in both, their types (i.e., between-touch setting) and their level of strength (i.e., within-touch setting). Such a comparative results also help verify the strength and shortcoming of different machine learning algorithms for touch classification. Third, it highlights some of the challenges whose solution concepts can pave the path for integration of touch sensors in such application domains as human–robot interaction (HRI).

## 2. Materials and Methods

### 2.1. Participants

Twenty-two younger adults (eleven females, age: M = 32.95, SD = 6.41) participated in our study. Data from four participants (three males and one female) were not recorded properly and therefore they were excluded from our analysis.

### 2.2. The Sensor

In our experiment, participants touched a mannequin that wore a touch-sensor vest. This vest consisted of two TactilusT M touch sensor mats manufactured by Sensor Products Inc. (Table 1). Each sensor mat had 16 × 16 sensor cells (i.e., input channels). We put the sensor mats on the front and the back of the mannequin’s upper body. The size of sensor mat (i.e., on each of the front and back sides) was 17.9 inch${}^{2}$. The size of a single sensor cell on each mat was 1.12 inch${}^{2}$. The sampling rate of the touch-sensor in this study was 50.0 Hz. At each recording step, the sensor vest generated, per front and back sides, a 32 × 32 data matrix whose entries corresponded to pressure intensity.

### 2.3. Data Selection

Our experiment included five types of touch scenarios. Each touch type had two levels of strength: gentle or strong. All participants followed the same order of touch type and strength. This resulted in ten (five types × two strengths) touches. They were:Hitting the mannequin’s chest gently/stronglyHugging the mannequin gently/stronglyHitting the mannequin’s shoulder gently/stronglyRubbing the mannequin’s shoulder gently/stronglyRubbing the mannequin’s chest gently/strongly

For all these touch scenarios, we first determined their duration in number of acquired frames. We noted that individuals’ touch durations varied substantially in their acquired number of frames (mean (M) = 600.71 frames, standard deviation (SD) = 115.33 frames, minimum (Min) = 180.00 frames, maximum (Max) = 950.00 frames). To use an equal number of frames and to also ensure retaining a maximum amount of data, we opted for the shortest number of frames (i.e., min = 180.00 frames) in our dataset. Given the touch sensor’s 50.0 Hz sampling rate, this translated to 3.60 s (i.e., from the onset of a session) of touch data, per participant, per touch scenario.

Next, we visually inspected the participants’ acquired data. For this purpose, we used an in-house Matlab 2016a script (Natick, MA, USA) that visualized the participants’ touch patterns on a contour (i.e., contourf function in Matlab) whose content was updated at real-time frame-by-frame. We further cross-validated these visualized touch patterns against the participants’ recorded videos during their touch sessions. Our inspection revealed that almost all participants actually performed “rubbing the mannequin’s shoulder strongly” as “hitting the mannequin’s shoulder strongly.” Therefore, we discarded the pair “rubbing the mannequin’s shoulder gently/strongly” from our study. During the inspection, we also realized that most participants’ data that was related to “hitting the mannequin’s shoulder strongly” did not have enough contrast when compared to their “rubbing the mannequin’s shoulder gently.” To acquire a higher resolution data, we computed the contrast ratio between “rubbing the mannequin’s shoulder strongly” (which was performed as “hitting the mannequin’s shoulder strongly” by most of the participants) and the actual “hitting the mannequin’s shoulder strongly,” per participant. We found that the latter yielded a higher overall ratio (i.e., $E[\frac{\mathrm{rubbing}\;\mathrm{the}\;\mathrm{mannequin}’s\;\mathrm{shoulder}\;\mathrm{strongly}}{\mathrm{hitting}\;\mathrm{the}\;\mathrm{mannequin}’s\;\mathrm{shoulder}\;\mathrm{strongly}}]>1$). Therefore, we replaced the participants’ “rubbing the mannequin’s shoulder strongly” with their respective “hitting the mannequin’s shoulder strongly” in the case of “hitting the mannequin’s shoulder gently/strongly” pair (i.e., scenario number 3 in the list above).

Further inspection of the participants’ data also revealed that not all participants touched the mannequin as intended (e.g., too soft to generate any tangible touch pattern on the touch sensor or performing additional undesired/unexpected touching of the mannequin that resulted in noisy data frames). In some cases, the mannequin also appeared not to be stable enough, thereby resulting in an inadequate touch data acquisition. As a result, we discarded all participants whose data were not well-defined (e.g., none or very few activated cells, mixed touch patterns, incorrectly performed touch, etc.). It is also worth noting that all scenarios included an excessive activation around the neck (i.e., irrespective of the touch scenario) as a result of the misalignment of the touch sensor on the mannequin’s torso. However, we did not filter out this unwanted activity since its removal would have caused loss of information and activation patterns in scenarios that included touching around this area (e.g., Rubbing/Hitting the mannequin’s chest/shoulder gently/strongly).

At the end of inspection, we kept all those touch scenarios whose pairs remained intact. However, the gentle/strong were not necessarily corresponded to the same participant in all cases. This resulted in final inclusion of six touch scenarios that corresponded to nine participants, per scenario. They were:Hitting the mannequin’s chest gently (Touch1)Hugging the mannequin gently (Touch2)Hitting the mannequin’s shoulder gently (Touch3)Hitting the mannequin’s chest strongly (Touch4)Hugging the mannequin’s strongly (Touch5)Hitting the mannequin’s shoulder strongly (Touch6)

Figure 1 shows examples of hitting the mannequin’s shoulder gently (i.e., Touch3, Figure 1A) and hugging the mannequin strongly (i.e., Touch5, Figure 1B). The two heatmaps in the bottom-left corner of each subplot show the pattern of sensor’s activation in response to their respective touch gestures. Comparing these heatmaps verifies that whereas Touch3 (i.e., hitting the mannequin’s shoulder gently) corresponded to weakly activated sensor’s cells, the values associated with the sensor’s cells activation in the case of Touch5 (i.e., hugging the mannequin strongly) exhibited more variability.

### 2.4. Analysis

#### 2.4.1. Touch Feature Computation

We utilized the 180 frames (i.e., 3.60 s), per participant, per touch scenario, and computed three types of features (for both gentle and stronger touch scenarios):Maximum Touch Activated Area (MTAA): We first found the frame with maximum number of activated touch sensor’s cells (i.e., out of 180 frames, per touch scenario, per participant). MTAA was then computed as m × 1.12${}^{2}$ where m refers to the number of activated cells and 1.12${}^{2}$ is the area of a single cell.Cumulative Sum of Touch Intensity (CSTI): This was calculated as the sum of activation (i.e., measured pressure, per cell) of all cells in the maximally touch activated frame that was used for computing the Maximally Activated Touch Area above.Relative Time of Maximally Activated Frame (RTMAF): This was computed as the index of the frame (i.e., out of 180 frames, per touch scenario) that corresponded to the frame with Maximally Activated Touch Area above.

#### 2.4.2. Touch Classification

We adapted six different classification methods to determine the utility of three extracted features for predicting the type and strength of different touch scenarios. These classifiers were random forest (RF), naive Bayes (NB), decision tree (DT), k-nearest-neighbor (KNN), support vector classifier (SVC), and logistic regression (LR).

To compare the performance of these classifiers, we performed 100 simulation runs in which we randomly split touch scenarios to 70.0% train and 30.0% test sets. We also ensured that a balanced proportion of each of the six labels were split between these train and test sets. We used the train set for training each of these classifiers and the test set to compute their prediction accuracy. We also determined whether including polynomial features improved their performance. For this purpose, we ran these algorithms while considering 0 (i.e., no polynomial feature) through 5 polynomial features. We observed that all of them yielded their best accuracy when they used two-degree polynomial features. Therefore, we reported the results based on two-degree polynomial features.

The participants’ calculated features formed inputs to these classifiers. We z-normalized these features. Figure 2 shows the z-normalized feature matrices for nine participants that were included in this study. The output from the classifiers was their respective predicted touch scenario (i.e., 1 through 6, one for each of the touch classes described in Section 2.3). Given the six different touch scenarios (i.e., three types × two strengths) that were balanced (i.e., all had equal number of samples = 9), the chance level accuracy was ≈16.67%.

We then used the 100 predictions by each of these algorithms and applied a Kruskal–Wallis test that was followed by posthoc Wilcoxon signed rank to determine the classifier with the highest accuracy. Our analyses indicated that RF significantly outperformed all the other classifiers. Therefore, we reported the RF’s results in the main manuscript. For RF performance, we reported its overall (i.e., averaged over 100 simulation runs) accuracy, precision, recall, and F1-score. We also provided its confusion matrix that showed the percentage of misclassification of each touch scenario with respect to the other touch scenarios.

#### 2.4.3. Improving the RF Accuracy

To determine whether RF high accuracy could be further improved, we applied two additional steps in our data preprocessing pipeline. These steps included the reduction of undesired persistent activity around the mannequin’s neck and the introduction of an additional feature using graph-based connected components. In what follows, we elaborate on these two steps:Reduction of Undesired Activity around a Mannequin’s Neck: In our data, we observed that the activity around the mannequin’s neck was present in all participants’ data and regardless of the touch gesture that they performed on this mannequin. Further investigation of these data revealed that such an undesired activity was present even prior to the start of the participants’ session. The latter observation verified that the observed noisy activity was due to the inadequate placement of the sensor vest on the mannequin’s upper body. To attenuate this effect, we extracted the sensor’s data of all the participants that pertained to the one frame prior to the start of their session, per touch scenario. Next, we located all the sensor’s cells that were commonly active in all of these frames (i.e., one per participant, per touch gesture). We then computed the MTAA and CSTI features (Section 2.4.2 and Figure 2) for this undesired activity around the mannequin’s neck. Subsequently, we subtracted them from all participants’ corresponding MTAA and CSTI features that were calculated during their sessions, per touch gesture.Introduction of an Additional Feature: We computed a new feature; largest connected component (LCC). In essence, LCC corresponded to the number of activated sensor’s cells in the largest connected area of the sensor vest. In this respect, whereas MTAA quantified the maximum number of activated sensor’s cells during a touch session, LCC represented the number of such cells that formed a connected neighboring cells that formed the largest subset of such pattern of activation. To compute LCC, we treated the sensor’s data in terms of a graph. Precisely, we first converted this data to an adjacency matrix by assigning a “1” or a “0” to every cell “c” if it was active or inactive:
$$\begin{array}{c} \left\{\begin{array}{cc} c_{ij}=1 & \mathrm{if}\;\mathrm{the}\;\mathrm{cell}\;\mathrm{at}\;\mathrm{row}\;i\;\mathrm{and}\;\mathrm{column}\;j\;\mathrm{was}\;\mathrm{active},\;\forall i,j=1,\cdots ,32\\ c_{ij}=0 & \mathrm{otherwise} \end{array}\right. \end{array}$$
where 32 represents a dimension of sensor’s frame (i.e., a 32 × 32 frame, Section 2.2) and c${}_{ij}$ refers to the sensor’s cell at i*^th^* row and j*^th^* column location on this frame. Next, we computed all the connected components [21] of this adjacency matrix. LCC was then the connected component that comprised the largest number of active cells among all of these connected components. Figure 3 shows the modified feature vectors, per touch gesture, per participant that included LCC.Figure 4 visualizes the pairwise cosine similarity distances between different touch gestures. This figure verifies that the use of [MTAA, CSTI, and RTMAF, LCC] feature vectors for quantification of these touch gestures quite effectively captured the similarity between participants’ data for each of these gestures. This can be seen by inspecting the nearly zero-valued larger-area squares, per touch pairs that lay along the diagonal. Looking at the larger-area squares along the row entries, these features were also able to capture considerable dissimilarities between touch gestures of different type. These observations indicated that [MTAA, CSTI, and RTMAF, LCC] feature vectors extracted substantial motion-related spatial information that were inherent characteristic/property of these touch gestures. This is due to the fact that the cosine similarity quantifies the similarities among a given set of vectors in terms of their directions in space.

#### 2.4.4. Statistics

For the Kruskal–Wallis test, we reported the effect size $r=\sqrt{\frac{\chi^{2}}{N}}$ [22] with *N* denoting the sample size and $\chi^{2}$ is its test-statistics. In the case of Wilcoxon test, we used $r=\frac{W}{\sqrt{N}}$ [23] as effect size with *W* denoting the Wilcoxon statistics and *N* being the sample size.

Data inspection, feature computation, and statistical tests were carried out in Matlab 2016a. For classifications, we used Python machine learning library, scikit-learn [24].

### 2.5. Ethics Statement

This study was carried out in accordance with the recommendation of the ethical committee of the Advanced Telecommunications Research Institute International (ATR) with written informed consent from all subjects. Every participant signed a written informed consent. The protocol was approved by the ATR ethical committee (approval code:17-601-4).

## 3. Results

### 3.1. Overall Accuracies

Table 2 summarizes the average (i.e., 100 simulation runs) overall (i.e., all touch gestures combined) accuracy (mean (M), standard deviation (SD)), precision, recall, and F1-score for RF, NB, DT, KNN, SVC, and LR. Considering these overall (i.e., all touch gestures combined) accuracies, we observed that all classifiers were able to achieve an above chance (i.e., ≈16.67%) performance. On other hand, we also observed a substantial variation among these classifiers’ performances. For instance, KNN and LR were associated considerably lower overall accuracies than other classifiers. Furthermore, RF and NB showed overall accuracies that were markedly higher than all other classifiers.

Kruskal–Wallis test indicated a significant difference between these accuracies with a large effect size (*p*< 0.00001, H(5, 599) = 334.93, r = 0.75). Posthoc Wilcoxon rank sum tests (Figure 5) further indicated significant differences in pairwise performance comparisons between these classifiers. Specifically, we observed that (Table 3) RF significantly outperformed all other classifiers’ performance and that such differences were associated with large effect sizes. The only exception to this observation was the significant difference between RF and NB in which we observed a medium effect size. Similarly, NB showed significantly higher accuracies than other classifiers. These significant differences were also associated with large effect sizes. Although we observed significant differences between DT, SVC, and KNN, these differences were more subtle than those observed in the case of RF and NB. In addition, these latter cases were all associated with small and medium effect sizes.

### 3.2. Between-Gesture Accuracies

We observed that all the classifiers showed above chance (≈16.67%) accuracies when their overall performance (i.e., all touch gestures combined) was considered. However, further analysis of these classifiers’ performance revealed a substantial variability when different touch gestures were taken into consideration (i.e., Touch1 through Touch6, Section 2.3).

Figure 6 shows the confusion matrix (left subplot) and the accuracy (right subplot) of RF, as per touch gesture. This figure indicates that RF achieved a relatively high accuracy in distinguishing between different touch gestures. This is in particular true in the case of Touch1 (i.e., Hitting the mannequin’s chest gently) and Touch5 (i.e., Hugging the mannequin’s strongly). This figure also shows that RF distinguished between the two strengths (i.e., gentle vs. strong) of each touch type relatively well. However, further inspection of this figure indicates that RF was not very successful in detecting differences among some of these settings. For instance, we observed that it misclassified Touch3 (i.e., Hitting the mannequin’s shoulder gently) and Touch6 (i.e., Hitting the mannequin’s shoulder strongly) and vice versa. Similarly, it confused Touch4 (i.e., Hitting the mannequin’s chest strongly) with Touch6 and Touch1. Worthy of note is also its misclassification of Touch2 (i.e., Hugging the mannequin gently) as Touch1.

In the case of NB (Figure 7), we also observed a relatively similar trend of high accuracy for predicting different touch gestures. However, its rate of misclassification was notably higher than the RF. This is in particular evident for Touch1, Touch3, Touch4, and Touch6.

On the other hand, such misclassifications were more substantial in the case of DT (Figure 8), KNN (Figure 9), SVC (Figure 10), and LR (Figure 11). This is in particular evident in the case of DT and KNN for Touch4 (Hitting the mannequin’s chest strongly) that showed close to (in the case of DT) or below (in the case of LR) chance accuracy for this touch gesture.

#### Improving the RF Accuracy

Figure 12 shows the RF performance after we included LCC in participants’ feature vectors (Section 2.4.3). This figure shows a substantial improvement in RF accuracy in the case of all touch gestures and regardless of their strength (i.e., gentle versus stronger touch). Wilcoxon signed rank test indicated a significant improvement in RF accuracy using (MTAA, CSTI, RTMAF, and LCC) (i.e., “after”) compared to its previous accuracy (Figure 13) using (MTAA, CSTI, and RTMAF) only (i.e., “before”) (*p* < 0.00001, W(198) = 9.06, r = 0.64, M${}_{After}$ = 96.67, SD${}_{After}$ = 3.37, M${}_{Before}$ = 85.00, SD${}_{Before}$ = 8.60). Appendix B provides the results for RF performance on an extended dataset that comprised nine touch gestures.

## 4. Discussion

The present study sought to realize the utility of machine learning approaches for classification of gentle vs. strong touch gestures. For this purpose, we used a mannequin and asked the human subjects to perform three types of touch gestures on its upper body. These gestures included hitting the mannequin’s chest, hugging it, and hitting its shoulder. We also asked our participants to perform these touch gestures in two different strengths: a gentle and a strong touch. This resulted in total of six different scenarios (i.e., three touch gestures × two strengths). The comparative analysis of machine learning approaches for touch gesture classification is not new and has been reported previously (e.g., Alonso-Martin et al. [18]). The present study complemented these previous results by adding another dimension to the problem of touch classification, namely, change in strength of the same touch gesture (i.e., gentle versus stronger hug).

We observed that a time window of ≈3.5 s was sufficient for extracting three features that quite reliably captured the essential information about the type and the strength of these touch gestures. These features were Maximally Activated Touch Area, Cumulative Sum of Touch Intensity, and Relative Time of Maximally Activated Frame. Our results implied that these features along with their interactions (in the form of polynomial degrees) could enable the RF classifier to achieve a relatively high accuracy on classification of touch types and their differing strengths (i.e., gentle versus stronger). Our results also suggested that this feature space could potentially help address issues concerning the touch interactions with shorter time duration [7] or the need for large number of features [16].

We observed that the inclusion of LCC in participants’ feature vectors could further increase their ability for quantification of the differences between touch gestures in terms of their spatial information. Precisely, we observed that the use of these features resulted in a relatively high discrimination between participants’ data for different touch gestures. This observation was evident in the cosine similarity between these feature vectors in which they appeared to quantify the dis/similarities between these touch gestures based on their implicit motion-based spatial information. This was due to the fact that the cosine similarity quantifies the similarities among the given set of vectors in terms of their direction in space. This, in turn, substantially improved the performance of RF classifier whose performance was already significantly higher than other classifiers that we used in our study (see Appendix B for RF performance on an extended dataset of nine touch gestures). Interestingly, the inclusion of LCC for capturing the spatial dynamics of these touch gestures appeared to be highly adequate for distinguishing between same touch gestures with different strengths (i.e., gentle versus stronger). In this respect, we observed that LCC significantly improved the RF performance on distinguishing between such different strengths compared to its performance when LCC was not included.

We also observed that almost all of the classifiers that were included in our analysis achieved above chance overall (i.e., all touch types combined) accuracies. Furthermore, these overall accuracies were based on separate test sets that were collected from different individuals and were balanced on their number of touch gestures, per touch type. In addition, these accuracies were obtained using a number of random split of data (100 simulation runs) between train and test sets. This differed from Braun et al. [16] whose results appeared to be solely based on the training set: a problem that is more commonly known as “circular analysis” or “double dipping” in various applications of machine learning algorithms [25]. The observation that almost all of these algorithms were able to achieve above chance level overall accuracies further implied that perhaps the use of more complex models such as artificial neural networks [7,14] might not be necessary if one solely relies on such overall accuracies.

On the other hand, these results more crucially underlined the danger of such interpretations as overall accuracy for drawing conclusion on the reliability of machine learning models. Specifically, although we observed that almost all of these classifiers achieved above chance overall (i.e., all touch types combined) accuracies, they rather exhibited significant differences in their performance per touch gesture. For instance, KNN and SVC mostly misclassified “Hitting the mannequin’s shoulder gently” (i.e., Touch3) as “Hitting the mannequin’s chest gently” (i.e., Touch1). Similarly, LR considerably misclassified Touch2 (i.e., “Hugging the mannequin gently”) as Touch1. This indicated that such strategies as distance-based similarity measures were insufficient for predicting different touch gestures. This observation held true whether such a similarity was measured based on distances between feature vectors of these touch gestures (e.g., KNN) or when more sophisticated optimization criterion was applied (e.g., maximization of their projection onto the decision boundary in the case of SVC). These results further verified that overall accuracy is not a reliable measure for drawing conclusion on the level of performance of different classifiers in such scenarios as touch interaction [9,10,11,12,13,14,17].

On the other hand, RF that relied on the shared information among features of these touch gestures appeared to yield significantly higher accuracy on predicting different touch types. In this regard, it is worth noting the remarks by Alonso-Martin et al. [18] who identified that the use of RF might result in overfitting. Although a more informed conclusion on this matter certainly require future investigation, we doubt that the observed RF performance in our study was due to overfitting. This is due to the following observations. First, we calculated our accuracies based on multiple random splits of data (i.e., 100 simulation runs). Second, the data in our dataset were collected from different individuals (i.e., non-overlapping samples) and comprised balanced (i.e., equal proportion of touch gestures) in each of these random test sets. Third, the RF’s confusion matrix (Figure 6) verified that its performance was not due to a subset of touch gestures/strengths (see Appendix A for analytical results).

Our results further identified that the performance of these models were even more variable once the strength of these touch gestures were taken into consideration. Precisely, we observed that, although some of these classifiers achieved above chance accuracies, their performance was only based on a subset of touch types. This was in particular evident in the case of KNN and DT. Interestingly, these two classifiers were substantially inaccurate for detecting the level of strength between touch gestures. Our results indicated that both DT and KNN mostly misclassified “Hitting the mannequin’s chest strongly” (i.e., Touch4) as “Hitting the mannequin’s chest gently” (i.e., Touch1) and “Hitting the mannequin’s shoulder strongly” (i.e., Touch6) as “Hitting the mannequin’s shoulder gently” (i.e., Touch3). Similarly, SVC misclassified Touch3 as Touch6 with the same rate and Touch4 as Touch1 with almost the same rate. Although we also observed such misclassifications in the case of RF, they were evidently lower in their occurrences. For instance, its misclassification of Touch4 as Touch1 accounted for 5.00% only. However, both RF and NB showed considerably high misclassification of Touch3 as Touch6 (30.0% and 20.0% in the case of NB and RF, respectively).

The higher specificity of RF in our results also signified the fact that a potentially appropriate solution to touch classification may not necessarily depend on more complex approaches (e.g., neural networks [7,14]) but the proper use of available methodologies. For instance, DT is the building block of RF: the latter is nothing but a collection of the former. Whereas the application of a single DT in our study resulted in a poor classification performance (particularly in the case of gentle versus stronger touch gestures), RT that built upon the consensus among collection of DTs achieved an evidently higher accuracy. This observation was also in line with Alonso-Martin et al. [18] in which LMT in their case outperformed such classifiers as convolutional neural network.

The above observations further highlighted the substantial impact of touch strength on its classification. This is certainly a critical point whose effect must be taken into consideration while using these classifiers in the context of automated touch recognition systems. The significance of this matter becomes more apparent once one considers the fact that the strength of a touch can transform its meaning. For instance, whereas a gentle squeeze might signal an affectionate treat, a stronger one could be construed as a sign of punishment by its recipient.

Our results hint at the potentials for the use of smart touch sensors in such application domains as socially assistive robotics (SAR) [6]. Such sensors can enable the robots to enhance the quality of their social interactions with their users by recognizing the type of these individuals’ touching patterns and behavior. They can also enrich their social services to these individuals by exhibiting reciprocal behaviors that complement such humans’ physical interactions [26,27].

## 5. Limitations and Future Directions

Although our results verified the utility of RF for classification of type and strength of three touch scenarios, it was the limitations and the shortcomings of our study that proved most informative. First, our results highlighted the pitfall of cramming experimental sessions with too many trials. Concretely, the preprocessing phase of our study showed that such settings can inadvertently become a source of confusion. For instance, we observed that our participants treated the “rubbing the mannequin’s shoulder strongly” as “hitting the mannequin’s shoulder strongly”. Even more striking was the observation that the latter showed a higher ratio with respect to their “hitting the mannequin’s shoulder gently”, thereby leading us to use it in place of the former.

Second, our study signified the challenge of the use of touch sensors in such naturalistic scenarios as touch interaction. For instance, we observed that a slight misalignment of the sensor could lead to such issues as undesired sensory information (e.g., unintended activity around the mannequin’s neck). Such a noise, in turn, had a detrimental effect in the form of loss of experimental trials in which participants’ actual touch trials were indistinguishable from this noise. Further implication of such a shortcoming came in the form of trial loss where we needed to discard an entire touch type. It is apparent that such a loss also meant further reduction of number of participants that were already limited in their numbers.

Although our further analysis enabled us to partially address the effect of sensor’s misalignment through identification of the noisy sensor area (i.e., the mannequin’s neck), we strongly believe that such temporary solution could not be sustained for long-term interaction scenarios. For instance, this noise was consistent in our case among all touch gestures and for all participants. We also had short sensor recordings for all participants before the start of their sessions. These allowed us to localize most sensor’s cells that were affected by its misalignment. On the other hand, such a fortunate case may not always be available. Therefore, thorough inspection and other precautionary steps to ensure the proper sensor’s alignment is a must. In the same vein, more research in the area of noise filtering and sensory data smoothing is necessary to devise more sophisticated tools for real-life physical interaction scenarios.

Another important issue that requires further investigation is the effect of adapted medium on study and analysis of touch-related interaction. For instance, the present study and the results that were presented by Silvera-Tawil et al. [11] both used mannequin as a medium. Whereas Silvera-Tawil et al. [11] used a touch sensor on a mannequin’s arm, we acquired touch data through a sensor vest that was worn on its upper body. This may suggest that the relatively flexible materials used in such touch sensors make them potentially applicable for the settings in which the interaction with more human-like media (e.g., humanoid robots) are envisioned. On the other hand, many robots that are used in socially-assistive scenarios are either pet-like robots [27] or may have body shapes that differ from that of humans [26]. In this regard, some of the previous results showed the potential utility of touch sensors in such application domains as pet-like robots [7] or media whose shapes very much differed from humans and other animals [9]. However, these results on their own do not necessarily promote their suitability to broader domains. Therefore, to further realize the utility of these results in such areas as socially-assistive robotics [6], future research must reevaluate them by considering robots of various forms. It is also crucial to validate the generalizability of these results by including individuals from wider age groups (e.g., children, older adults), cultural backgrounds (e.g., international studies), and differential neuropsychological capacities (neurotypical versus more neurodiverse individuals such as autistic children).

These limitations and issues can become even more overwhelming when one considers their potential impacts on determining the type and the strength of a touch during a real-life naturalistic scenario. Specifically, real-life settings will involve interacting agents in which both parties may re/act to the received touch. Furthermore, and unlike a mannequin, these agents will have their own body movements, thereby deflecting the sensor’s surface which can, in turn, lead to patterns of activation that have nothing to do with the actual interaction.

In this context, the use of other modalities and their potential fusion [18] to increase the accuracy of classifiers is an exciting venue that deserves further investigation. Such systems can endow socially-assistive agents with the capacity for building more realistic models of human physical interactions that are highly complex and context/cultural-dependent [3,4,5].

Taken together, our results highlight the possibility of detecting humans’ touch with differing type/strength. They also (perhaps more importantly) pinpoint the necessity for more focused experimental paradigms that will include small but fundamental types of touch along with larger number of participants, thereby allowing for more informed conclusions on this line of research.

## Figures and Tables

**Figure 1 sensors-20-03033-f001:**
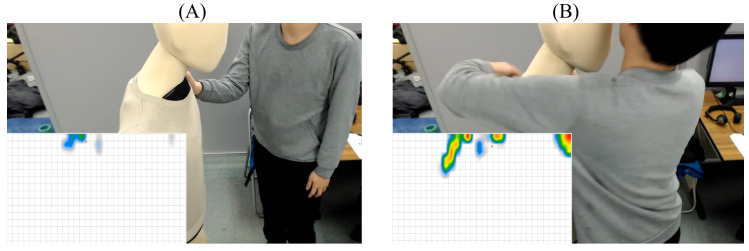
(**A**) Hitting the mannequin’s shoulder gently (**B**) Hugging the mannequin strongly. In these figures, the heatmaps visualize the sensor values with red and blue indicating strength and gentle touches. These figures verify that whereas Touch3 (i.e., hitting the mannequin’s shoulder gently) corresponded to weakly activated sensor’s cells, the values associated with the sensors’ cells activation in the case of Touch5 (i.e., hugging the mannequin strongly) exhibited more variability.

**Figure 2 sensors-20-03033-f002:**
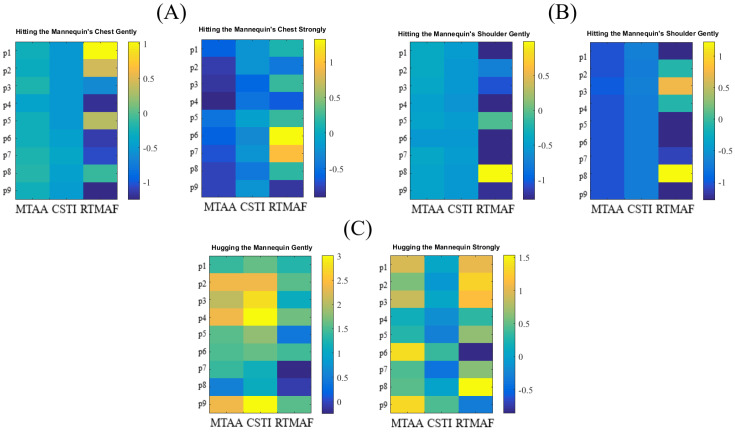
Touch Features for (**A**) Hitting the mannequin’s chest gently (left) and strongly (right) (**B**) Hugging the mannequin gently (left) and strongly (right) (**C**) Hitting the mannequin’s shoulder gently (left) and strongly (right). The calculated features are shown along the *x*-axis. They are Maximally Touch Activated Area (MTAA), Cumulative Sum of Touch Intensity (CSTI), and Relative Time of Maximally Activated Frame (RTMAF). The values depicted in these subplots are z-normalized (i.e., each column is mean-subtracted and divided by its standard deviation). The *y*-axis corresponds to the participants in this study (i.e., p1 through p9).

**Figure 3 sensors-20-03033-f003:**
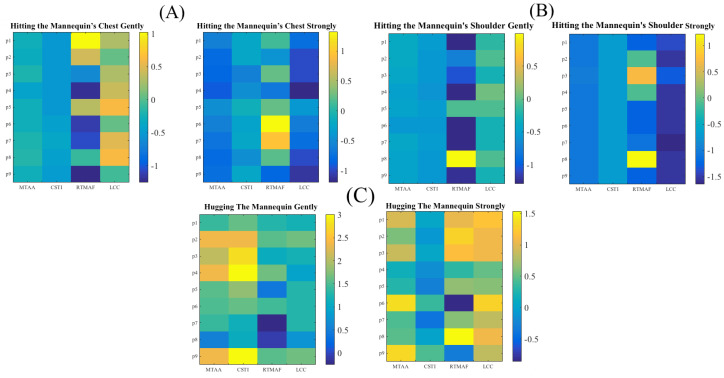
Modified Touch Features. (**A**) Hitting the mannequin’s chest gently (left) and strongly (right); (**B**) Hugging the mannequin gently (left) and strongly (right); and (**C**) Hitting the mannequin’s shoulder gently (left) and strongly (right). MTAA, CSTI, RTMAF, and LCC features are shown along the *x*-axis. In these subplots, MTAA, CSTI, and RTMAF are same as in Figure 2. The values in these subplots are z-normalized (i.e., each column is mean-subtracted and divided by its standard deviation). The *y*-axis corresponds to the participants in this study (i.e., p1 through p9).

**Figure 4 sensors-20-03033-f004:**
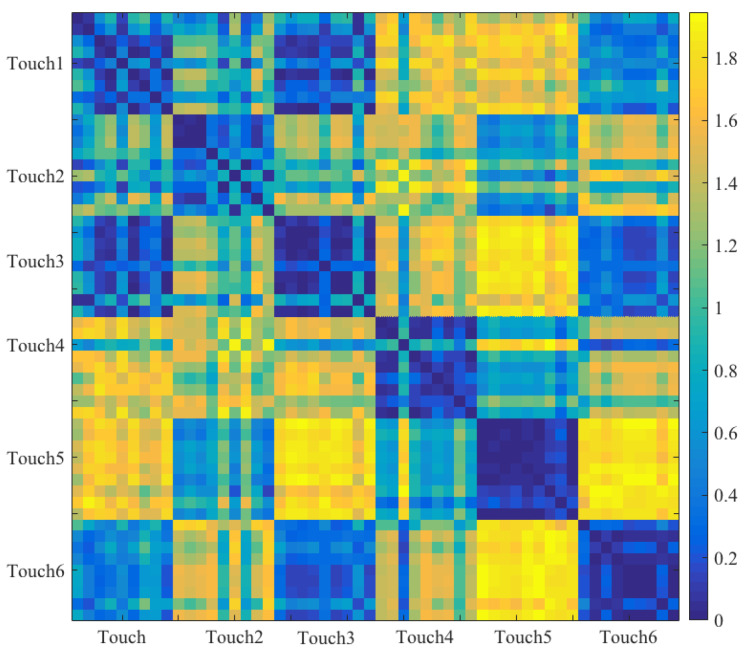
Pairwise cosine similarity distance between all touch gestures’ feature vectors. This figure verifies that MTAA, CSTI, RTMAF, and LCC quite effectively captured the similarity between participants’ data that pertained to same touch gestures. This is evident in the nearly zero-valued larger-area squares, per touch pairs that lay along the diagonal. Looking at the larger-area squares along the row entries, per touch pairs, these features were also able to extract considerable dissimilarities between gestures of different type. Each of these larger-area squares is a 9 × 9 matrix: i.e., one small square (a cosine similarity) per participant and every other participant (i.e., self-similarities included). As a result, each touch gesture is associated with a 9 × 54 (i.e., 9 participants × 6 touch gestures) cosine similarity sub-matrix that is extended along the row of the overall cosine similarity matrix of all touch gestures in this figure.

**Figure 5 sensors-20-03033-f005:**
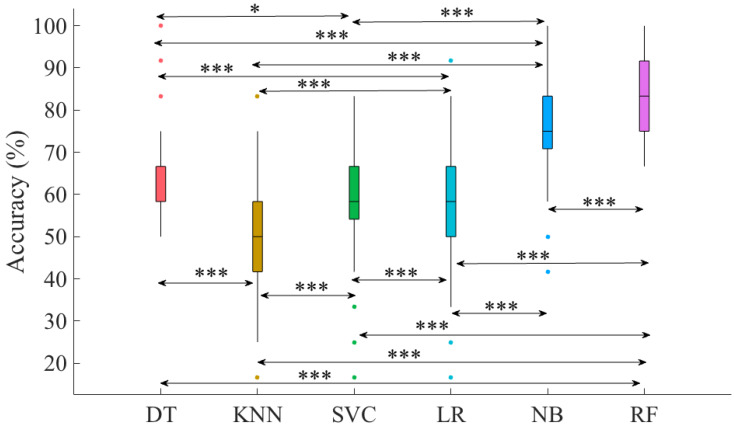
Models’ overall accuracies. Asterisks mark the significant differences between these classifiers’ performance as indicated by their pairwise Wilcoxon rank sum tests (*: *p* < 0.01, ***: *p*< 0.00001). In this figure, significantly higher overall accuracy (chance level 16.67%) of RF compared to other classifiers is evident.

**Figure 6 sensors-20-03033-f006:**
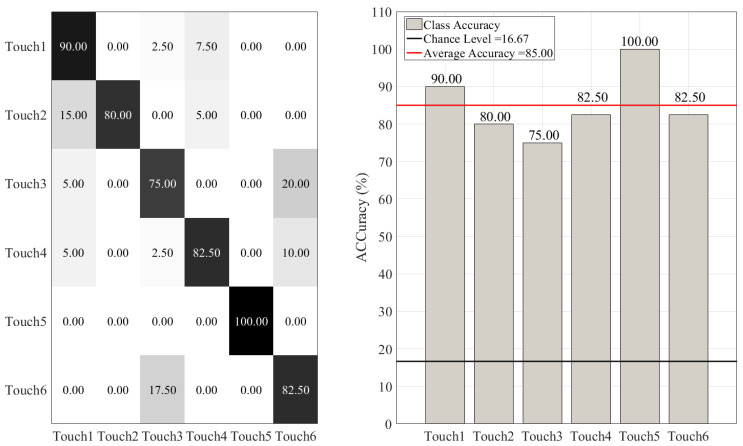
Random forest (RF) classifier. The subplot on the right shows the accuracy of this classifier for each touch scenario (i.e., Touch1 through Touch6). The subplot on the left presents the confusion matrix for this classifier. The overall accuracy of random forest (red-line in the right subplot) was 85.00% (chance level (black-line in the right subplot) ≈ 16.67, given six-class classification of balanced classes).

**Figure 7 sensors-20-03033-f007:**
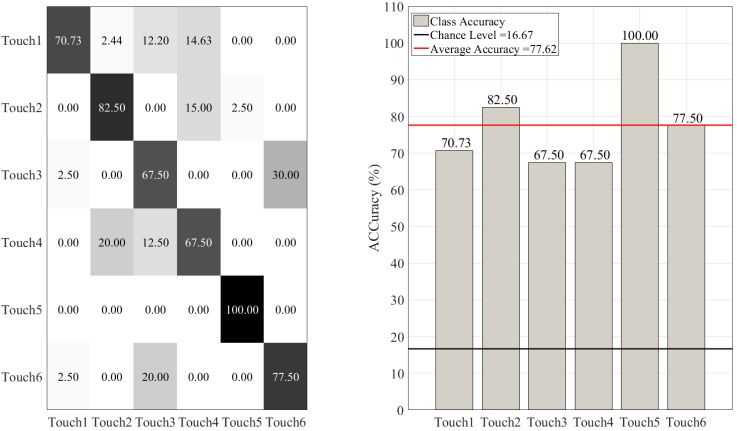
Naive Bayes (NB) classifier. The subplot on the right shows the accuracy of this classifier for each touch scenarios (i.e., Touch1 through Touch6). The subplot on the left presents the confusion matrix for this classifier. The overall accuracy of Naive Bayes (red-line in the right subplot) was 77.62% (chance level (black-line in the right subplot) ≈ 16.67, given six-class classification of balanced classes).

**Figure 8 sensors-20-03033-f008:**
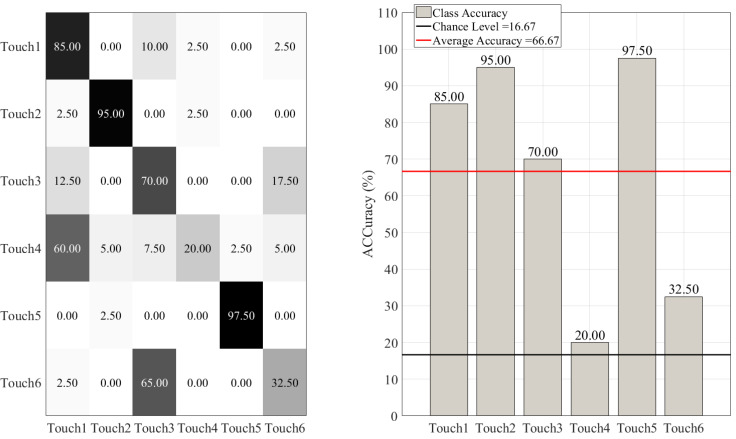
Decision tree (DT) classifier. The subplot on the right shows the accuracy of this classifier for each touch scenarios (i.e., Touch1 through Touch6). The subplot on the left presents the confusion matrix for this classifier. The overall accuracy of decision tree (red-line in the right subplot) was 66.67% (chance level (black-line in the right subplot) ≈ 16.67, given six-class classification of balanced classes).

**Figure 9 sensors-20-03033-f009:**
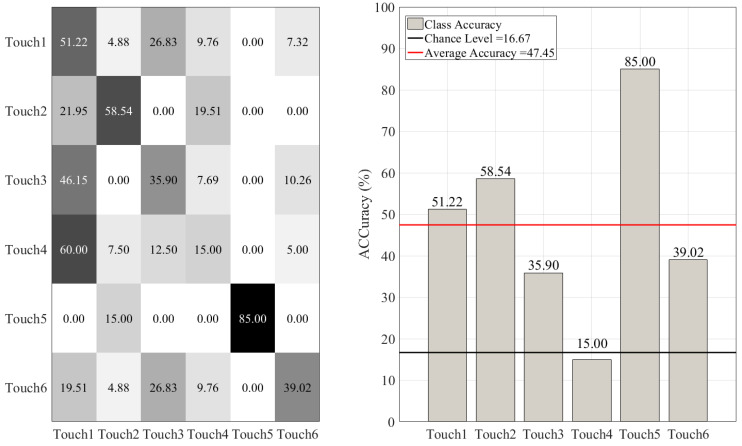
K-nearest-neighbor (KNN) classifier. The subplot on the right shows the accuracy of this classifier for each touch scenarios (i.e., Touch1 through Touch6). The subplot on the left presents the confusion matrix for this classifier. The overall accuracy of K-nearest-neighbor (red-line in the right subplot) was 47.45% (chance level (black-line in the right subplot) ≈ 16.67, given six-class classification of balanced classes).

**Figure 10 sensors-20-03033-f010:**
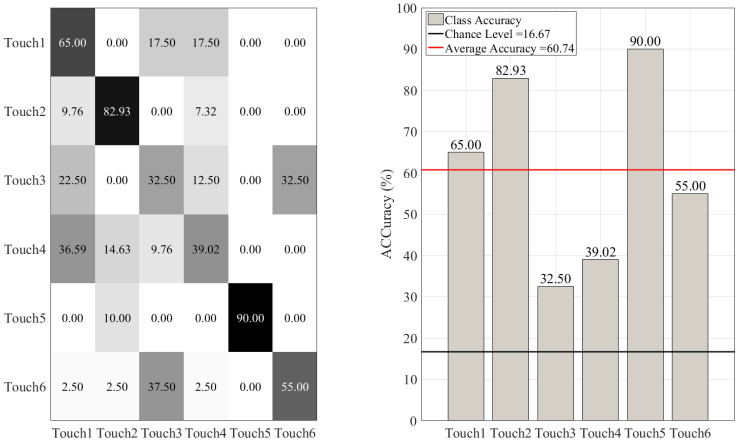
Support vector classifier (SVC) with radial basis kernel. The subplot on the right shows the accuracy of this classifier for each touch scenarios (i.e., Touch1 through Touch6). The subplot on the left presents the confusion matrix for this classifier. The overall accuracy of support vector classifier (red-line in the right subplot) was 60.74% (chance level (black-line in the right subplot) ≈ 16.67, given six-class classification of balanced classes).

**Figure 11 sensors-20-03033-f011:**
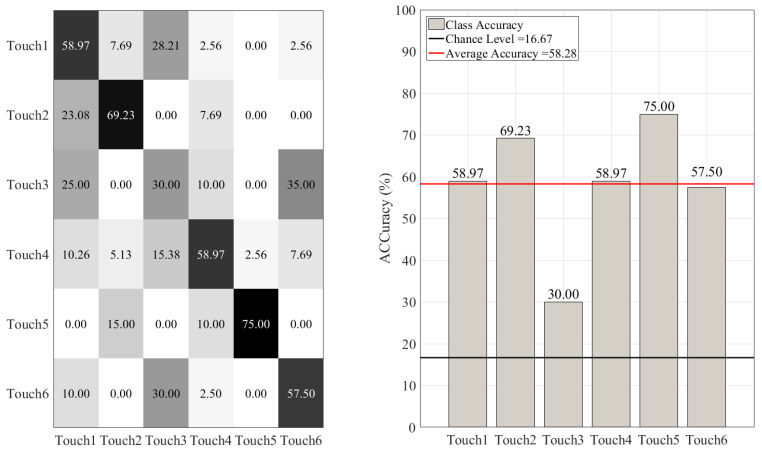
Logistic regression (LR) classifier. The subplot on the right shows the accuracy of this classifier for each touch scenarios (i.e., Touch1 through Touch6). The subplot on the left presents the confusion matrix for this classifier. The overall accuracy of logistic regression (red-line in the right subplot) was 58.28% (chance level (black-line in the right subplot) ≈ 16.67, given six-class classification of balanced classes).

**Figure 12 sensors-20-03033-f012:**
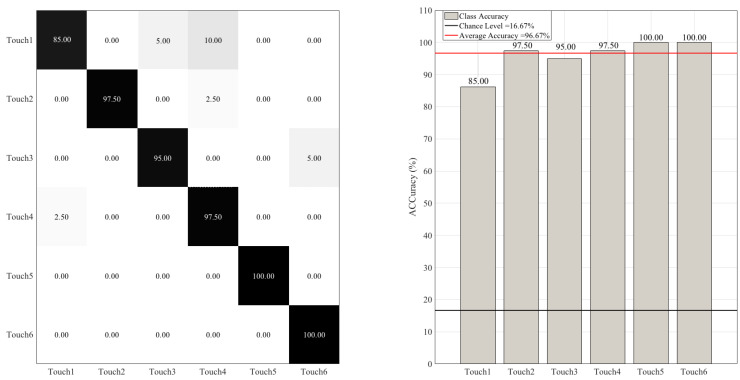
Random forest (RF) classifier. The subplot on the right shows the accuracy of this classifier for each touch scenarios (i.e., Touch1 through Touch6). The subplot on the left presents the confusion matrix for this classifier. The overall accuracy of random forest (red-line in the right subplot) was 85.00% (chance level (black-line in the right subplot) ≈ 16.67, given six-class classification of balanced classes).

**Figure 13 sensors-20-03033-f013:**
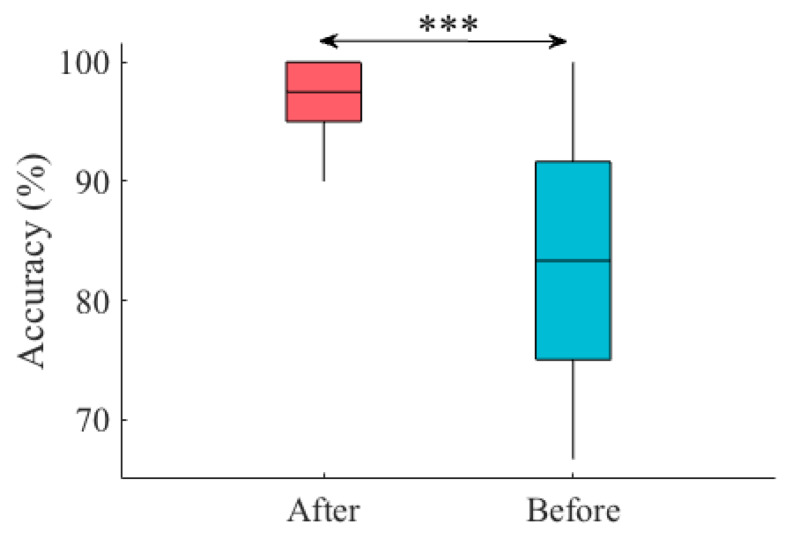
RF overall accuracy “before” and “after” original feature vectors (i.e., (MTAA, CSTI, RTMAF, LCC)) were extended by six additional features that were the Euclidean distances between the averaged cosine similarities (Section 2.4.3). Asterisks mark the significant differences between these classifiers’ performance as indicated by their pairwise Wilcoxon rank sum tests (***: *p*< 0.00001).

**Table 1 sensors-20-03033-t001:** Specifications of the Sensor Products Inc. used in the present study.

Technology	Piezoresistive
Pressure Range	0.1 to 200 PSI (0.007 to 14.1 kg/cm^2^)
Matrix Size	Up to 64 × 256 lines
Thickness	From 12 mils (0.3 mm)
Mat Sensor Size	Customizable up to 150" (381 cm)
1Scan Speed	Up to 1000 hertz
Min Sensing Point Size	0.188 in2 (1.21 cm^2^)
Stretchability	Up to 158%
Accuracy	±10%
Repeatability	±2%
Hysteresis	±5%
Nonlinearity	±1.5%
Calibration	NIST Traceable

**Table 2 sensors-20-03033-t002:** Overall (i.e., all touch gestures combined) average (i.e., 100 simulation runs) accuracy, precision, recall, and F1-score associated with random forest (RF), naive Bayes (NB), decision tree (DT), k-nearest-neighbor (KNN), support vector classifier (SVC), and logistic regression (LR). M and SD stand for the mean and the standard deviation of each model’s accuracies in 100 simulation runs.

Classifier	Accuracy	Precision	Recall	F1-Score
RandomForest	M = 85.00%, SD = 8.60	0.89	0.87	0.86
Naive Bayes	M = 77.62 %, SD = 10.65	0.86	0.83	0.82
DecisionTree	M = 66.67%, SD = 9.66	0.68	0.64	0.66
KNN	M = 47.45%, SD = 12.90%	0.57	0.54	0.53
Support Vector	M = 60.74%, SD = 12.68%	0.60	0.62	0.61
Logistic Regression	M = 58.28%, SD = 14.47	0.57	0.56	0.58

**Table 3 sensors-20-03033-t003:** Paired posthoc Wilcoxon rank sum tests between average (i.e., 100 simulation runs) accuracy of every pairs of models. The models are: random forest (RF), K-nearest-neighbor (KNN), support vector classifier (SVC), logistic regression (LR), Naive Bayes (NB), and decision tree (DT). W and r refer to the Wilcoxon test-statistics and effect size, respectively. Significantly superior performance of RF compared to all other models is evident in this table.

Models	*p* <	W(198)	r
RF vs. KNN	0.00001	12.12	0.86
RF vs. SVC	0.00001	10.86	0.77
RF vs. LR	0.00001	11.06	0.78
RF vs. NB	0.00001	4.54	0.32
RF vs. DT	0.00001	10.08	0.71
NB vs. KNN	0.00001	11.33	0.80
NB vs. LR	0.00001	9.04	0.64
NB vs. SVC	0.00001	8.42	0.60
NB vs. DT	0.00001	7.07	0.50
KNN vs. SVC	0.00001	−7.07	−0.50
KNN vs. LR	0.00001	−5.06	−0.36
KNN vs. DT	0.00001	−9.67	−0.68
SVC vs. LR	0.00001	7.07	0.50
SVC vs. DT	0.01	−3.02	−0.21

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
