# Peer review of "Gentle Versus Strong Touch Classification: Preliminary Results, Challenges, and Potentials"

_sensors, 2020, doi:10.3390/s20113033_

Round 1
Reviewer 1 Report
This paper focuses on sought to realize the utility of machine learning approaches for classification of gentle vs. strong touch gestures. Comparing with previous study results, this study added another dimension to the problem of touch classification, namely, change in strength of the same touch gesture (i.e., gentle versus stronger hug). This study adapted six different classification methods ( random forest (RF), naive Bayes (NB), decision tree (DT), k-nearest-neighbour (KNN), support vector classifier (SVC), and logistic regression (LR)) to determine the utility of three extracted features (Maximally Activated Touch Area, Cumulative Sum of Touch Intensity and Relative Time of Maximally Activated Frame) for predicting the type and strength of different touch scenarios. However, the reviewer believes that this manuscript should be improved in the following areas: 1. As mentioned in this paper ('2.3. Data Selection' and '5. Limitations and Future Direction'), slight misalignment of the sensor could lead to such issues as undesired sensory information. And in the actual interaction, this problem is difficult to overcome. It is expected that the author can introduce this influencing factor into the classifier as a negative input parameter, increase the number of effective samples to obtain more accurate experimental results. 2. The remarks by Alonso-Martin et al. [19] identified that the use of RF might result in overfitting. In '4. Discussion', author mentioned the RF performance in our study may be due to overfitting, but only mentioned three observations factor. It is expected that the author can analysis and verify in detail whether RF performance in this study was overfitting in this study, make the experimental results of this article more convincing. 3. In the 'Abstract', 'In fact, touch classification literature represents an undeniably progressive results.' It seems that 'results' need change to 'result'. 4. In 'Table 2', 'Paired posthoc Wilcoxon rank sum tests between average (i.e., 100 simulation runs) accuracy of every every pairs of models.' It seems that 'every' needs to be deleted. 5. In the third paragraph of '4. Discussion', 'This differed from Braun et al. [17] whose results appeared to be solely based on the training set: a problem that is more commonly known as "circular analysis" or "double dipping" in various application of machine learning algorithms [26].' It seems that 'application' needs change to 'applications'. 6. In the third paragraph of '5. Limitations and Future Direction', 'In this regards, some of the previous results showed the potential utility of touch sensors in such application domains as pet-like robots [22] or media whose shapes very much differred from humans and other animals [10].' It seems that ' differred' need change to ' differed'.
Author Response
First and foremost, the authors would like to take this opportunity to thank the reviewer for the time spent and the kind consideration to review our manuscript. The comments by the reviewer helped us improve the quality of our results and their presentation substantially.
In what follows, we provide our responses to the reviewer’s comments and concerns.
Sincerely,
Reviewer 1
Reviewer’s Comment: 1. As mentioned in this paper ('2.3. Data Selection' and '5. Limitations and Future Direction'), slight misalignment of the sensor could lead to such issues as undesired sensory information. And in the actual interaction, this problem is difficult to overcome. It is expected that the author can introduce this influencing factor into the classifier as a negative input parameter, increase the number of effective samples to obtain more accurate experimental results.
Authors’ Response: We addressed the reviewer’s concern in two steps: solving for the influencing factor and increasing the sample.
-
Solving for the influencing factor:We added a new Section (2.4.3. Improving The RFAccuracy,lines 217-248in the current version of the manuscript).In this Section, we first tackled the issue of undesirable persistent sensor’s activation around mannequin’s neck. Specifically, since this activation was due to misalignment of the sensor vest on the mannequin’s upper body, it was present in all participants’ data, per touch session. Therefore, we used the sensor’s data right before the start of their touch session (i.e., last frame before experiment started, per participant, per touch session) for all participants and all touch sessions. We then found all sensor’s cells that were active and were common among all of these frames. For this common area, we computed MTAA and CSTIfeatures (please refer to Section 2.4.1. Touch Feature Computation, lines 180-189, points 1 and 3for their description)andsubtracted themfrom the participants’ MTAA and CSTI that were computed for their touch experiments. This step allowed to us attenuate the effect of these commonly activated area on MTAA and CSTI values for such touch gestures as hitting the shoulder/chest that were very close to the noisy mannequin’s body part (i.e., its neck), thereby (at least partially) increasing the signal-to-noise ratio for all touch gestures.
In the current version of the manuscript,we provided this explanation under the heading “1. Reduction of Undesired Activity Around Mannequin’s Neck”,lines221-232in Section 2.4.3. Improving The RFAccuracy.
We then opted for acquiring more information about each touch gesture’s characteristics and their differences through finding more informative features. We achieved this objective by making use of “connected component” from graph theory and cosine similarity distance between touch gestures’ features. In the case of connected component, we treated the selected sensor’s activated frame as a graph by first transforming this frame into an adjacency matrix. Specifically, we assigned a “1” to every activated cell and a “0” to those cells that were not activated. This step transformed the frame to undirected graph in which every cell with a value “1” represented an edge. We then computed all connected components associated with this adjacency matrix and then selected the one with largest number of elements (i.e., activated cells). We called this the largest connected component (LCC) and used it as a feature. At this point, every touch gesture, per participant, was represented by a feature vector of length 4 (i.e., [MTAA CSTI RTMAF LCC]). It is worth noting that LCC differed fromMTAA in that whereas MTAA represented the total count of all active cells, LCC captured the largest activated area whose cell were connected (i.e., formed by continuously neighbouring cells) and therefore formed the largest area of the sensor that was affected by the touch. In the current version of the manuscript,we provided this explanation under the heading “2.Introduction of AnAdditional Feature”,lines233-248in Section 2.4.3. Improving The RFAccuracy.We visualized the each of the steps in Figures 3 and 4, page 8.
We reported the results of RF algorithm (i.e., the classifier that showed the significantly highest accuracy compared to other classifiers in our study) based on these two stepsin Section 3.2.1. Improving The RFAccuracy, lines 303-309and Figures 12 and 13, page 15.
We further discussed these new results in Section 4. Discussion, lines330-34.It reads as follows.
“We observed that the inclusion of LCC in participants’ feature vectors could further increase their ability for quantification of the differences between touch gestures in terms of their spatial information. Precisely, we observed that the use of these features resulted in a relatively high discrimination between participants’ data for different touch gestures. This observation was evident in the cosine similarity between these feature vectors in which they appeared to quantify the dis/similarities between these touch gestures based on their implicit motion-based spatial information. This was due to the fact that the cosine similarity quantifies the similarities among the given set of vectors in terms of their direction in space. This, in turn, substantially improved the performance of RF classifier whose performance was already significantly higher than other classifiers that we used in our study (see Appendix B for RF performance on an extended dataset of nine touch gestures). Interestingly, the inclusion of LCC for capturing the spatial dynamics of these touch gestures appeared to be highly adequate for distinguishing between same touch gestures with different strengths (i.e., gentle versus stronger). In this respect, we observed that LCC significantly improved the RF performance on distinguishing between such different strengths compared to its performance when LCC was not included.”
We also elaborated on the shortcoming associated with these new results in 5. Limitations and Future Direction, lines 428-447.
“Although our further analysis enabled us to partially address the effect of sensor’s misalignmentthrough identification of the noisy sensor area (i.e., the mannequin’s neck), we strongly believe that such temporary solution could not be sustained for long-term interaction scenarios. For instance, this noise was consistent in our case among all touch gestures and for all participants. We also had short sensor recordings for all participants before the start of their sessions. These allowed us to localize most sensor’s cells that were affected by its misalignment. On the other hand, such a fortunate case may not always be available. Therefore, thorough inspection and other precautionary steps to ensure the proper sensor’s alignment is a must. In the same vein, more research in the area of noise filtering and sensory data smoothing is necessary to devise more sophisticated tools for real-life physical interaction scenarios.”
-
Increasing the sample: We applied our approach to attenuation of the undesired sensor’s activation around the mannequin’s neck (“1. Reduction of Undesired Activity Around Mannequin’s Neck”,lines221-232in Section 2.4.3. Improving The RFAccuracy) and recovered three our four remaining touch gestures that were excluded from the first version of our manuscript. These three touch gestures were “Rubbing the mannequin’s shoulder gently (Touch7),” “Rubbing the mannequin’s chest strongly (Touch8),” and “Rubbing the mannequin’s chest gently (Touch9).” It is also worth noting that we were not able to use the touch gesture "Rubbing the mannequin’s shoulder strongly" since our earlier inspection revealed that almost all participants performed "rubbing the mannequin’s shoulder strongly" as "hitting the mannequin’s shoulder strongly." (please see Section 2.3for details).
We reported the results of RF algorithm (i.e., the classifier that showed the significantly highest accuracy compared to other classifiers in our study) on this extended data in Appendix B. RF Performance Using a Larger Number of Touch Gestures (lines502-526,in the current version of the manuscript)along with Figures A.2 and A.3,page 21.
Reviewer’s Comment: 2. The remarks by Alonso-Martin et al. [19] identified that the use of RF might result in overfitting. In '4. Discussion', author mentioned the RF performance in our study may be due to overfitting, but only mentioned three observations factor. It is expected that the author can analysis and verify in detail whether RF performance in this study was overfitting in this study, make the experimental results of this article more convincing.
Authors’ Response: In the current version of manuscript, we added a newAppendix(Appendix A Verification of the Non-Overfitting Performance of Random Forest (RF))that presents analytical results for non-overfitting performance of RF. Specifically, we provided two-step analysis: a qualitative step and a quantitative step. In the case of qualitative analysis, we presented the learning curve of our model which shows how RF’s accuracy on test data consistently increased as a function of training samples. This provided a qualitative evidence that RF performance was not due to a potential overfitting. In this respect, the overfitting could have been identified if the RF accuracy on test was associated with an inverted bowl-shape curve. In other words, if RF initially would have shown an improved performance and then its accuracy on test would have started to decline. On the other hand, considering its test accuracy curve that consistently reduced its distance with its training curve (i.e., they were getting closer) as a function of training sample indicated that RF’s learning was improved by more information available during its model’s training.
In the case of quantitative evaluation, we performed two sets of bootstrap (1000 simulation runs) test of significance at 95.0% confidence interval (CI95.0%).First,we considered the overall (i.e., all touch gestures combined) RF accuracy. Next,we applied this test on each of the touch gestures separately. Specifically, for each of these testswe computed the bootstrapped accuracy of RF and then subtracted the average performance of RF from these bootstrapped estimations. In this context, the RF accuracy could have been due to the presence of outliers (and therefore possibility of overfitting during those simulation runs) if zero was not contained within the confidence interval of the difference between average RF accuracy and its bootstrapped estimated accuracy.
In the current version of the manuscript, we provided this information in lines 477-500(i.e., Appendix A Verification of the Non-Overfitting Performance of Random Forest (RF)).We further visualized our results inFigure A1, page19,and summarized the results of the bootstrap tests in Table A1, page 20.We referred to this appendix in Section 4. Discussion, line 377.
Reviewer’s Comment: 3. In the 'Abstract', 'In fact, touch classification literature represents an undeniably progressive results.' It seems that 'results' need change to 'result'.
Authors’ Response: We changed “results” to “result”
Reviewer’s Comment: 4. In 'Table 2', 'Paired posthoc Wilcoxon rank sum tests between average (i.e., 100 simulation runs) accuracy of every every pairs of models.' It seems that 'every' needs to be deleted.
Authors’ Response: We deleted the extra “every” from the text.
Reviewer’s Comment: 5. In the third paragraph of '4. Discussion', 'This differed from Braun et al. [17] whose results appeared to be solely based on the training set: a problem that is more commonly known as "circular analysis" or "double dipping" in various application of machine learning algorithms [26].' It seems that 'application' needs change to 'applications'.
Authors’ Response: We changed “application” to “applications”
Reviewer’s Comment: 6. In the third paragraph of '5. Limitations and Future Direction', 'In this regards, some of the previous results showed the potential utility of touch sensors in such application domains as pet-like robots [22] or media whose shapes very much differred from humans and other animals [10].' It seems that ' differred' need change to ' differed'.
Authors’ Response: We corrected “differred”.

Reviewer 2 Report
The article is very well written. The motivation, experiments, results and conclusions are described well. The only advice I would offer to the authors is to provide more detail about the tactile sensing apparatus itself. Other than that, the authors did a good job on this article.
Author Response
First and foremost, the authors would like to take this opportunity to thank the reviewer for the time spent and the kind consideration to review our manuscript. The comments by the reviewer helped us improve the quality of our results and their presentation substantially.
In what follows, we provide our responses to the reviewer’s comments and concerns.
Sincerely,
Reviewer 2
The article is very well written. The motivation, experiments, results and conclusions are described well. The only advice I would offer to the authors is to provide more detail about the tactile sensing apparatus itself. Other than that, the authors did a good job on this article.
Authors’ Response: The authors thank the reviewer for the encouraging words. In the current version of the manuscript, we added a table (Table 1, Section The Sensor, page 4) that provides the device specifications.

Reviewer 3 Report
This manuscript has studied and compared the performance of several classifiers in the classification of touch gestures. The author placed the touch sensor vest on the mannequin and let the experimenter touch the mannequin. The author designed a variety of different touch scenes for experimenters to perform, so as to obtain enough data for the classifier to perform the classification test. The classification results of the classifier are analyzed and compared in detail, and the advantages and disadvantages of each classifier are summarized. The RF classifier is proved to be a better classifier in the touch gesture classification. This research provides a solution for detecting different types/intensities of touch in machine learning. I suggest that the paper can be published after appropriate modification:
- The experimental data of the participants excluded in 2.1 should be written in the paper, and the result will be more real.
- Ten different touch scenes were designed and implemented in 2.3, but only six touch scenes are provided in the following text. If all the experimental results are written into the paper, the content of the paper will be more abundant.
- The author did not compare the experimental results of this paper with the experimental results of other papers. It is recommended to add charts for comparison to reflect the advantages of this paper.
Author Response
First and foremost, the authors would like to take this opportunity to thank the reviewer for the time spent and the kind consideration to review our manuscript. The comments by the reviewer helped us improve the quality of our results and their presentation substantially.
In what follows, we provide our responses to the reviewer’s comments and concerns.
Sincerely,
Reviewer 3
Reviewer’s Comment:1. The experimental data of the participants excluded in 2.1 should be written in the paper, and the result will be more real.
Authors’ Response: We addressed the reviewer’s concern in two steps: solving for the influencing factor and increasing the sample.
-
Solving for the influencing factor: We added a new Section (2.4.3. Improving The RFAccuracy,lines 217-248in the current version of the manuscript).In this Section, we first tackled the issue of undesirable persistent sensor’s activation around mannequin’s neck. Specifically, since this activation was due to misalignment of the sensor vest on the mannequin’s upper body, it was present in all participants’ data, per touch session. Therefore, we used the sensor’s data right before the start of their touch session (i.e., last frame before experiment started, per participant, per touch session) for all participants and all touch sessions. We then found all sensor’s cells that were active and were common among all of these frames. For this common area, we computed MTAA and CSTI features (please refer to Section 2.4.1. Touch Feature Computation, lines 180-189, points 1 and 3for their description)and subtracted them from the participants’ MTAA and CSTI that were computed for their touch experiments. This step allowed to us attenuate the effect of these commonly activated area on MTAA and CSTI values for such touch gestures as hitting the shoulder/chest that were very close to the noisy mannequin’s body part (i.e., its neck), thereby (at least partially) increasing the signal-to-noise ratio for all touch gestures.
In the current version of the manuscript,we provided this explanation under the heading “1. Reduction of Undesired Activity Around Mannequin’s Neck”,lines221-232in Section 2.4.3. Improving The RFAccuracy.
We then opted for acquiring more information about each touch gesture’s characteristics and their differences through finding more informative features. We achieved this objective by making use of “connected component” from graph theory and cosine similarity distance between touch gestures’ features. In the case of connected component, we treated the selected sensor’s activated frame as a graph by first transforming this frame into an adjacency matrix. Specifically, we assigned a “1” to every activated cell and a “0” to those cells that were not activated. This step transformed the frame to undirected graph in which every cell with a value “1” represented an edge. We then computed all connected components associated with this adjacency matrix and then selected the one with largest number of elements (i.e., activated cells). We called this the largest connected component (LCC) and used it as a feature. At this point, every touch gesture, per participant, was represented by a feature vector of length 4 (i.e., [MTAA CSTI RTMAF LCC]). It is worth noting that LCC differed fromMTAA in that whereas MTAA represented the total count of all active cells, LCC captured the largest activated area whose cell were connected (i.e., formed by continuously neighbouring cells) and therefore formed the largest area of the sensor that was affected by the touch. In the current version of the manuscript,we provided this explanation under the heading “2.Introduction of AnAdditional Feature”,lines233-248in Section 2.4.3. Improving The RFAccuracy.We visualized the each of the steps in Figures 3 and 4, page 8.
We reported the results of RF algorithm (i.e., the classifier that showed the significantly highest accuracy compared to other classifiers in our study) based on these two stepsin Section 3.2.1. Improving The RFAccuracy, lines 303-309and Figures 12 and 13, page 15.
We further discussed these new results in Section 4. Discussion, lines330-34.It reads as follows.
“We observed that the inclusion of LCC in participants’ feature vectors could further increase their ability for quantification of the differences between touch gestures in terms of their spatial information. Precisely, we observed that the use of these features resulted in a relatively high discrimination between participants’ data for different touch gestures. This observation was evident in the cosine similarity between these feature vectors in which they appeared to quantify the dis/similarities between these touch gestures based on their implicit motion-based spatial information. This was due to the fact that the cosine similarity quantifies the similarities among the given set of vectors in terms of their direction in space. This, in turn, substantially improved the performance of RF classifier whose performance was already significantly higher than other classifiers that we used in our study (see Appendix B for RF performance on an extended dataset of nine touch gestures). Interestingly, the inclusion of LCC for capturing the spatial dynamics of these touch gestures appeared to be highly adequate for distinguishing between same touch gestures with different strengths (i.e., gentle versus stronger). In this respect, we observed that LCC significantly improved the RF performance on distinguishing between such different strengths compared to its performance when LCC was not included.”
We also elaborated on the shortcoming associated with these new results in 5. Limitations and Future Direction, lines 428-447.
“Although our further analysis enabled us to partially address the effect of sensor’s misalignment tthrough identification of the noisy sensor area (i.e., the mannequin’s neck), we strongly believe that such temporary solution could not be sustained for long-term interaction scenarios. For instance, this noise was consistent in our case among all touch gestures and for all participants. We also had short sensor recordings for all participants before the start of their sessions. These allowed us to localize most sensor’s cells that were affected by its misalignment. On the other hand, such a fortunate case may not always be available. Therefore, thorough inspection and other precautionary steps to ensure the proper sensor’s alignment is a must. In the same vein, more research in the area of noise filtering and sensory data smoothing is necessary to devise more sophisticated tools for real-life physical interaction scenarios.”
-
Increasing the sample: We applied our approach to attenuation of the undesired sensor’s activation around the mannequin’s neck (“1. Reduction of Undesired Activity Around Mannequin’s Neck”,lines221-232in Section 2.4.3. Improving The RFAccuracy) and recovered three our four remaining touch gestures that were excluded from the first version of our manuscript. These three touch gestures were “Rubbing the mannequin’s shoulder gently (Touch7),” “Rubbing the mannequin’s chest strongly (Touch8),” and “Rubbing the mannequin’s chest gently (Touch9).” It is also worth noting that we were not able to use the touch gesture "Rubbing the mannequin’s shoulder strongly" since our earlier inspection revealed that almost all participants performed "rubbing the mannequin’s shoulder strongly" as "hitting the mannequin’s shoulder strongly." (please see Section 2.3 for details).
We reported the results of RF algorithm (i.e., the classifier that showed the significantly highest accuracy compared to other classifiers in our study) on this extended data in Appendix B. RF Performance Using a Larger Number of Touch Gestures (lines 502-526, in the current version of the manuscript) along with Figures A.2 and A.3, page 21.
Reviewer’s Comment:2. Ten different touch scenes were designed and implemented in 2.3, but only six touch scenes are provided in the following text. If all the experimental results are written into the paper, the content of the paper will be more abundant.
Authors’ Response: please see our response to the reviewer’s comment “Reviewer’s Comment: 1. The experimental data of the participants excluded...”
Reviewer’s Comment:3. The author did not compare the experimental results of this paper with the experimental results of other papers. It is recommended to add charts for comparison to reflect the advantages of this paper.
Authors’ Response: We agree with reviewer’s comment on the importance of such comparison. In fact, we followedthispractice in the past (e.g.,[1]).However, in those cases we had access to data that was used with all other publications that used the same data, thereby making a direction comparison between our results and those of other published results plausible.
In the present study, the lack of similar data didn’t allow us for a direct comparative analysis of our results and the other. In fact doing so would not have born any significant/meaningful insights since our data differed from the other published results.Specifically, most results in the literature report the overall accuracy of their models. We highlighted this matter in Section 1. Introduction, lines 66-80. Another issue for applying direct comparison among our results and the others is with respect to scenarios in which touch strength (e.g., gentle versus stronger) varies. To the best of our knowledge, no previous study focused on this setting. We discussed this in Section 1. Introduction, lines 81-91.Inthis regards, one of the main goal of our study was to determine the effect of touch strength (e.g., gentle versus stronger) could bear any impact on their classification. We clarified this point in Abstract, line 11-14and Section 1. Introduction, lines 95-100.
However, we partially addressed this inability for direct comparison with other studies’ results by performingourcomparative analyses usingdifferent classifiers that were used in the previous publications. In fact, we chose these classifiers based on their use cases in previous studies. We clarified this point in the Abstract, line 12,as well as Section 1. Introduction, line 95-100.
[1]Soheil Keshmiri, Hidenobu Sumioka, Junya Nakanishi, and Hiroshi Ishiguro, Emotional State Estimation Using a Modified Gradient-Based Neural Architecture with Weighted Estimates, International Joint Conference on Neural Netowrks (IJCNN'17), Anchorage, USA 2017
